# Qualitative and Quantitative Analyses of Sialyl *O*-Glycans in Milk-Derived Sialylglycopeptide Concentrate

**DOI:** 10.3390/foods13172792

**Published:** 2024-09-02

**Authors:** Junichi Higuchi, Masaki Kurogochi, Toshiyuki Yamaguchi, Noriki Fujio, Sho Mitsuduka, Yuko Ishida, Hirofumi Fukudome, Noriko Nonoyama, Masayuki Gota, Mamoru Mizuno, Fumihiko Sakai

**Affiliations:** 1Milk Science Research Institute, Megmilk Snow Brand Co., Ltd., 1-1-2 Minamidai, Kawagoe-shi 350-1165, Saitama, Japan; junichi-higuchi@meg-snow.com (J.H.); toshiyuki-yamaguchi@meg-snow.com (T.Y.); noriki-fujio@meg-snow.com (N.F.); s-mitsuduka@meg-snow.com (S.M.); y-ishida@meg-snow.com (Y.I.); h-fuku@meg-snow.com (H.F.); noriko-adachi@meg-snow.com (N.N.); m-gota@meg-snow.com (M.G.); 2Laboratory of Glyco-Organic Chemistry, The Noguchi Institute, 1-9-7 Kaga, Itabashi-ku, Tokyo 173-0003, Japan or kurogochi.masaki.d2@f.mail.nagoya-u.ac.jp (M.K.); mmizuno@noguchi.or.jp (M.M.)

**Keywords:** sialylglycopeptide, *O*-glycan, homogeneous *O*-glycopeptide

## Abstract

Sialyl glycans have several biological functions. We have previously reported on the preparation and bifidogenic activity of milk-derived sialylglycopeptide (MSGP) concentrate containing sialyl *O*-glycans. The current study qualitatively and quantitatively analyzed the sialyl *O*-glycans present in the MSGP concentrate. Notably, our quantitative analysis indicated that a majority of *O*-glycopeptides in the MSGP concentrate were derived from glycomacropeptides. The concentrate was found to contain mainly three types of sialyl core 1 *O*-glycans, with the disialyl core 1 *O*-glycan being the most abundant. We successfully quantified three types of sialyl core 1 *O*-glycans using a meticulous method that used homogeneous *O*-glycopeptides as calibration standards. Our results provide valuable insights into assessment strategies for the quality control of *O*-glycans in dietary products and underscore the potential applications of MSGP concentrate in the food industry and other industries.

## 1. Introduction

Milk, a rich source of nutritionally valuable protein, offers a promising avenue for health research. Its proteins, which boast a well-balanced amino acid profile, excellent digestibility, and numerous bioactive peptides (antioxidant, antihypertensive, immunomodulatory, and antimicrobial peptides), have been the focus of recent proteomics studies [1,2]. In fact, such studies have revealed that various milk proteins, including lactoferrin, α-lactalbumin, immunoglobulin, κ-casein, and several milk fat globule membrane proteins, are extensively glycosylated [3,4,5]. The glycan moieties of these glycoproteins play crucial roles in protein folding, biological recognition, and protection from digestion [6]. They have also been shown to protect infants against pathogens [7,8] and serve as substrates for bifidobacterial growth in the infant’s gut [9,10]. The potential of the glycan moiety of milk glycoproteins to become a new functional material for maintaining and improving health inspires hope for the future of nutrition research. In particular, sialic acid attached to the non-reducing ends of these glycopeptides has shown potential in promoting brain development, immunomodulation, and even stress and inflammation relief [11,12].

Glycomacropeptide (GMP), which is a hydrophilic peptide released into whey from κ-casein during the production of cheese, is one of the most heavily glycosylated peptides in whey proteins. Studies over the past decades have characterized the *O*-glycan structures and the site-specific profiling of *O*-glycosylation in GMP. A total of eleven distinct glycan structures have been identified in bovine GMP. Of these, five are present in both colostrum and mature milk, whereas the remaining six are exclusively found in colostrum [6,13,14]. Moreover, Kurogochi et al. reported *O*-acetylation of sialic acid in GMP [5]. A total of seven potential sites of *O*-glycosylation in GMP have been reported, including ^121^T, ^131^T, ^133^T, ^136^T, ^141^S, ^142^T, and ^165^T [15,16]. In a recent study, intact GMP in commercial GMP powders was identified through a top-down approach based on liquid chromatography (LC)–mass spectrometry (MS) analysis [15,17]. The glycosylation of intact GMP and the presence of fragment GMP were identified in commercial GMP powders [17]. These glycans, especially Neu5Ac, are of great importance in the biological activities of GMP, including the inhibition of influenza virus binding to oligosaccharide receptors on host cells [18], the promotion of *Bifidobacterium* growth [19], and the facilitation of early brain development in young piglets [20].

In our previous study, we developed a unique “milk-derived sialylglycopeptide (MSGP)” concentrate from GMP-rich whey protein concentrate (G-WPC) [21]. The glycan types and structures in the MSGP concentrate, prepared from G-WPC, were consistent with those previously reported for GMP, mainly sialyl core 1 *O*-glycans. The concentrate has been demonstrated to possess higher bifidogenic properties than those of GMP in vitro. Given that no food ingredient has ever utilized sialyl *O*-glycan, the MSGP concentrate represents a potential game changer in the field of nutraceuticals and functional foods. We anticipate that the MSGP concentrate will be prepared from any cheese whey-derived whey protein concentrate (WPC) or whey protein isolate (WPI) containing GMP. It should be noted, however, that the *O*-glycans present in WPC or WPI are not limited to those derived from GMP. For quality control, when materials other than G-WPC are used as raw materials, it is necessary to analyze the chemical structure and quantity of sialyl *O*-glycans in the concentrate.

To accurately identify the structure of a glycopeptide, the glycosylation sites on the peptide chain need to be determined in addition to conducting a structural analysis of the peptide chain and glycopeptide. Unlike *N*-glycosylation, which features a consensus motif (NXS/T), no consensus motif exists for *O*-glycosylation. The use of LC–MS/MS with collision-induced dissociation (CID) and electron transfer dissociation (ETD) is indispensable when identifying the glycosylation sites of *O*-glycopeptides contained in the MSGP concentrate [5], underscoring the need for advanced techniques in our research.

*O*-glycopeptides are significantly more difficult to quantify than *N*-glycopeptides. Although *N*-glycans attached to asparagine residues can be released from proteins or peptides by commercially available enzymes, such as peptide-*N*-glycosidase F (PNGase F), and then identified or quantified through LC–MS techniques [22], no enzyme can release intact *O*-glycans attached to threonine or serine residues. *O*-glycans can be released from proteins or peptides via β-elimination under mild alkaline conditions; however, some of the released glycans are degraded under alkaline conditions, referred to as peeling reaction, thereby complicating the structural analysis and quantification of *O*-glycans. Kameyama et al. developed eliminative oximation that enables β-elimination of *O*-glycans from glycoproteins using hydroxylamine and 1,8-diazabicyclo[5.4.0]undec-7-ene (DBU) to suppress *O*-glycan degradation [23]. *O*-glycans released following this procedure were then analyzed using LC with a fluorescence detector (LC–FLD) after labeling with 2-aminobenzoate (2-AA). Yamaguchi et al. relatively quantified *O*-glycans in human milk using LC–MS without labeling, taking advantage of the fact that the *N*-acetylgalactosamine (GalNAc) at the reducing end of the *O*-glycan immediately forms an oxime upon release from the glycoprotein using Kameyama’s method; the oximated *O*-glycans are to be more ionized by electrospray ionization than native *O*-glycans [24]. These advanced methods are valuable for identifying and quantifying *O*-glycans in MSGP concentrate.

The current study aimed to qualitatively and quantitatively characterize the sialyl *O*-glycans in an MSGP concentrate prepared from whey protein isolates. We demonstrated that most of the *O*-glycopeptides in the MSGP concentrate were derived from GMP. To quantify *O*-glycans, we purified homogeneous *O*-glycopeptides from MSGP concentrate to be used as standards for calibration and then successfully quantified three types of sialyl core 1 *O*-glycans. These meticulous methods ensure the accuracy and reliability of our findings.

## 2. Materials and Methods

### 2.1. Reagents

Whey protein isolate (WPI) from cheese whey was purchased from a commercial manufacturer (Glanbia Nutritionals, Twin Falls, ID, USA). Alcalase 2.4 L FG (EC 3.4.21.62, 2.4 Anson unit/g, produced by *Bacillus licheniformis*, with endoprotease activity) and Flavorzyme 1000 L (EC 3.4.11.1, 1000 leucine amino-peptidase unit/g, produced by *Aspergillus oryzae*, with both endoprotease and exopeptidase activities) were obtained from Novozymes (Copenhagen, Denmark). The EZGlyco O-glycan Prep Kit was purchased from Sumitomo Bakelite (Tokyo, Japan). Other reagents were purchased from standard vendors.

### 2.2. Preparation of the MSGP Concentrate from WPI

The MSGP concentrate was prepared from WPI according to previously reported methods [21] with slight modifications. Briefly, WPI dissolved in water (10% [wt/wt]) was heated to 55 °C and adjusted to pH 7.0 using a KOH solution. Alcalase and Flavorzyme (0.45% [wt/wt], respectively) were then added to the solution simultaneously. Enzyme reaction was performed at 55 °C for 8 h and stopped by heating to 85 °C. We used a molecular weight cutoff 1000 ultrafiltration membrane (XT-3B-3838, Synder Filtration, CA, USA) to concentrate the glycopeptides from the WPI hydrolysate. The hydrolysate was then concentrated through ultrafiltration. The retentate was lyophilized using a freeze dryer (Nissei Limited, Tokyo, Japan). The resulting lyophilized powder, that is, the MSGP concentrate, was stored at −30 °C until further use.

### 2.3. Gross Chemical Composition of WPI and the MSGP Concentrate

The moisture, protein, fat, and ash contents of the MSGP concentrate were determined at the Japan Food Research Laboratories (Tokyo, Japan). Moisture content was determined using the oven-drying method under normal pressure. Protein content was determined using the Kjeldahl method with a nitrogen-to-protein conversion factor of 6.38. Fat content was determined using the Rose Gottlieb method. Ash content was determined using the dry ashing method. Carbohydrate content was calculated by subtracting the moisture, protein, fat, and ash contents from the total weight.

### 2.4. Determination of Sialic Acid and Galacto-N-Biose

To measure sialic acid (Neu5Ac) and galacto-*N*-biose (GNB), WPI and MSGP concentrates were treated with neuraminidase derived from *Arthrobacter ureafaciens* (EC 3.2.1.18, one unit is defined as the amount of enzyme required to liberate 1 μmol of Neu5Ac per minute at pH 5.0 at 37 °C, Nacalai Tesque Inc., Kyoto, Japan) and *O*-glycosidase (EC 3.2.1.97, one unit is defined as the amount of enzyme required to liberate 0.68 nmol of *O*-linked disaccharides from 5 mg of neuraminidase-treated fetuin in 1 h at 37 °C, New England Biolabs, MA, USA). A reaction mixture containing 5 mg/mL of WPI or 500 μg/mL of the MSGP concentrate, 0.6 U/mL of neuraminidase, and 800,000 U/mL of *O*-glycosidase in 50 mmol/L sodium phosphate buffer (pH 5.0) was incubated at 37 °C for 2 h. The Neu5Ac and GNB released by the enzymes were analyzed using high-performance anion-exchange chromatography with pulsed amperometric detection (HPAE-PAD). Enzymatic hydrolysates were applied to a Dionex CarboPac PA1 column (4 mm × 250 mm; 10 μm particle size; Thermo Fisher Scientific, Waltham, MA, USA). Standards for Neu5Ac and GNB were obtained as the purest available grades (Sigma-Aldrich, St. Louis, MO, USA).

For Neu5Ac analysis, the mobile phase consisted of water (solvent A), 200 mmol/L NaOH solution (solvent B), and 600 mmol/L sodium acetate in 100 mmol/L NaOH solution (solvent C) with the following gradient elution: from 0 to 10 min 45% B and 10% C; from 10 to 25 min 37.5% B and 25% C; and from 25 to 30 min 45% B and 10% C at a flow rate of 1 mL/min. For GNB analysis, the mobile phase consisted of solvents A, B, and C with the following gradient elution: from 0 to 9 min 50% B and 0% C; from 9 to 12 min 0% B and 100% C; and from 12 to 30 min 50% B and 0% C at a flow rate of 1 mL/min. The chromatograms were analyzed using Chromeleon software version 7.3 (Thermo Fisher Scientific).

### 2.5. Enzymatic Digestion of the MSGP Concentrate

A solution of MSGP concentrate (5 mg/mL) in 50 mmol/L phosphate buffer (pH 5.0) was treated with 1.2 units/mL of neuraminidase from *Arthrobacter ureafaciens* and 1,600,000 units/mL of *O*-glycosidase at 37 °C for 2 h. Subsequently, an equal amount of 2 U/mL of Proteinase K (EC 3.4.21.64, one unit is defined as the amount of enzyme required to produce a peptide equivalent to 1 μmol/min of tyrosine as the colorant of the Folin & Ciocalteu phenol reagent, Fujifilm Wako Pure Chemical Corporation, Osaka, Japan) was added to the sample followed by treatment at 37 °C for 1 h.

### 2.6. Size-Exclusion Chromatography (SEC)

To estimate the molecular weight of WPI, MSGP, and enzyme-digested MSGP, SEC analysis was performed using the Waters Alliance e2695 Separations Module (Waters Corporation, Milford, MA, USA). Samples were applied to a tandem combination of Inertsil Diol (4.6 mm × 250 mm; 5 μm particle size; GL Sciences, Inc., Tokyo, Japan) and Inertsil WP300 Diol (4.6 mm × 250 mm; 5 μm particle size; GL Sciences, Inc.) columns and eluted with a mobile phase consisting of 40% (vol/vol) acetonitrile containing 0.01% (vol/vol) trifluoroacetic acid at a flow rate of 0.3 mL/min. Absorbance was monitored at 210 nm using a 2489 UV detector (Waters Corporation). β-Lactoglobulin (molecular weight: 18,277), α-lactoalbumin (molecular weight: 14,146), aprotinin (molecular weight: 6512), sialylglycopeptide (from egg yolk, molecular weight: 2866), and oxytocin (molecular weight: 1007) were used as molecular weight markers.

### 2.7. Analysis of O-Glycan Composition in the MSGP Concentrate

Labeled MSGP *O*-glycans were prepared using an EZGlyco O-glycan Prep Kit. Briefly, 10 μL of MSGP solution (10 mg/mL in H_2_O) was mixed with 5 μL of Glycan Released Reagent A and 10 μL of Glycan Released Reagent B and incubated at 50 °C for 20 min. Subsequently, released glycans were captured using Glycan Capturing Beads, which were washed with acetonitrile. Thereafter, 4 mg of 2-aminobenzeamide (2-AB) and 0.04 mg of reducing reagent in 50 μL of methanol/acetic acid/H_2_O (9/2/9) were added to the beads. The *O*-glycan-containing solution was recovered via centrifugation at 3000× *g* for 1 min. The solution was incubated at 50 °C for 2.5 h and washed with acetonitrile to remove excess reagent with a Cleanup Column. The 2-AB-labeled *O*-glycans were recovered by adding H_2_O and analyzed using a Q-Exactive mass-spectrometer and an RS Fluorescence (FL) Detector coupled with an UltiMate 3000 (Thermo Fisher Scientific). For analysis, a labeled *O*-glycan solution (10 μL) was applied to a Glycanpac AXH-1 column (2.1 mm × 150 mm; 3 μm particle size; Thermo Fisher Scientific) at 40 °C. The mobile phase comprised acetonitrile (solvent A) and 50 mmol/L ammonium formate (pH 4.4) (solvent B) in a gradient elution of from 0 to 55 min at 10–35% and then from 55 to 65 min at 35% B. The flow rate was 0.4 mL/min. The electrospray voltage and heat capillary temperature were 3.5 kV and 275 °C, respectively. Nitrogen (99.5% purity) was used as sheath gas (set to 35), auxiliary gas (set to 10), and collision gas. Full-scan mass spectra were acquired in positive ion mode from *m*/*z* 400 to 2000 and a resolution of 70,000. MS2 spectra were acquired in the automatic data-dependent mode using one precursor scan, followed by five MS2 scans with a higher energy collisional dissociation. The stepped normalized collision energies were set to 10, 15, and 20. FL detector excitation and emission wavelengths were set to 330 and 430 nm, respectively. The spectra and chromatograms were analyzed using Xcalibur version 4.0 (Thermo Fisher Scientific). The structures of labeled *O*-glycans were identified through manual inspection.

### 2.8. Analysis of O-Glycopeptides in the MSGP Concentrate Using LC ESI-MS and ESI-MS/MS

LC ESI-MS and ESI-MS/MS were used to analyze the MSGP concentrate on an LC system (Dionex, Sunnyvale, CA, USA) with a C-18 reverse phase column (Inertsustain ODS-3, 1.0 × 150 mm, 3.5 μm particle size; GL Sciences, Inc.). The sample was loaded with 25 mmol/L ammonium formate solution (0–4 min), eluted with a gradient of 0–15% acetonitrile with 0.1% formic acid (4–34 min), washed with 100% acetonitrile with 0.1% formic acid (34–40 min), and then equilibrated with 25 mmol/L ammonium formate solution (40–60 min) at a constant flow of 50 μm/min (column oven at 37 °C). Electrospray MS data were collected using the HESI-II probe ion source on an LTQ-Velos Pro instrument (Thermo Fisher Scientific) in the positive mode. The spray voltage was set at 3.5 kV, with a temperature of 150 °C. The sheath gas flow and auxiliary gas flow rates were set at 20 and 5 arb, respectively. MS/MS data were collected using a data-dependent acquisition method, which was implemented as a “top 15” experiment: one precursor scan covering an *m/z* range of 400–2000, followed by CID spectra targeting the top 15 most intense ions in the precursor spectrum. CID event parameters were as follows: isolation width, 3 Da; normalized collision energy, 30%. The peaks and masses were integrated using the annotation procedures (Qual Browser) on Xcalibur 2.2 SP 1.48 software (Thermo Fisher Scientific).

The CID fragmentation normalized energy of the selective ion was set to 30%. ETD fragmentation for *O*-glycopeptides was performed using supplemental collisional activation (35%). MS3 measurements (CID-MS3) were performed through CID fragmentation of peptide-only ions (those with complete loss of the sugar chain from the CID-MS/MS data of *O*-glycopeptides), and the second CID fragmentation normalized energy was set to 35%.

### 2.9. Preparation of Homogeneous O-Glycopeptides as Standards for Quantification

*O*-glycopeptides were separated using a preparative HPLC system (PLC761; GL Sciences Inc., Tokyo, Japan) equipped with a UV detector (210 nm) and fraction collector on a HILIC amino column (NH2P-90 20F, 20.0 × 300 mm, 9 μm particle size; Resonac, New York, NY, USA) or a C18 RP column (YMC-Pack ODS-A, 20 × 150 mm, 5 μm particle size; YMC Co., LTD, Kyoto, Japan).

For the separation of *O*-glycopeptide with a disialyl core 1 *O*-glycan, a 5 mL solution of the MSGP concentrate (10 mg/mL) was loaded onto a HILIC amino column equilibrated with 30 mmol/L NaH_2_PO_4_ solution. Elution was performed with 30 mmol/L (0–5 min), a gradient of 30–165 mM (5–35 min), and 165–300 mM (35–45 min) NaH_2_PO_4_ solution at a flow of 10.0 mL/min at 40 °C. Each fraction was collected every 40 s. Fractions containing *O*-glycopeptide with a disialyl core 1 *O*-glycan were desalted using a graphite carbon cartridge (InertSep GC column; GL Sciences). The *O*-glycopeptide was eluted with 50% acetonitrile with 0.1% formic acid from the cartridge. The acetonitrile was then removed using a centrifugal evaporator (CVE-3100D; Tokyo Rikakikai Co., Ltd., Tokyo, Japan) after neutralization by adding a 28% ammonia solution. The desalted sample was lyophilized as a semi-purified *O*-glycopeptide with a disialyl core 1 *O*-glycan using EYELA FDU-2200 (Tokyo Rikakikai Co., Ltd.). To obtain a homogeneous *O*-glycopeptide with a disialyl core 1 *O*-glycan, a sample containing the semi-purified *O*-glycopeptide with a disialyl core 1 *O*-glycan dissolved in water was loaded onto a C18 RP column equilibrated with 2.0% acetonitrile with 0.1% formic acid. Elution was performed with a gradient of 2.0–10.8% (0–15 min) and 10.8–90.0% (15–25 min) and an isocratic mode of 90% (25–30 min) at a flow of 10.0 mL/min at 40 °C. Each fraction was collected every 30 s. Fractions containing *O*-glycopeptide with a disialyl core 1 *O*-glycan were desalted and lyophilized as described above. The lyophilized sample was used as the standard for *O*-glycopeptide with a disialyl core 1 *O*-glycan.

To prepare *O*-glycopeptide with a branched monosialyl core 1 *O*-glycan, 10 mL of homogeneous *O*-glycopeptide having a disialyl core 1 *O*-glycan (2 mg/mL) in 50 mmol/L of sodium phosphate buffer (pH 6.0) was treated with five units of α2,3-neuraminidase (EC 3.2.1.18, from *Salmonella typhimurium* LT2; Takara Bio Inc., Shiga, Japan) at 37 °C for 2 h to selectively release one molecule of Neu5Ac from *O*-glycopeptide with a disialyl core 1 *O*-glycan. Subsequently, α2,3-neuraminidase was inactivated through treatment at 100 °C for 5 min. The sample treated with α2,3-neuraminidase was loaded onto a HILIC amino column equilibrated with 15 mmol/L NaH_2_PO_4_. Elution was performed with 15 mmol/L NaH_2_PO_4_ (0–5 min) and with a gradient of 15–75 mmol/L (5–35 min), 75–150 mmol/L (35–45 min), and 150 mmol/L (45–50 min) at a constant flow of 10.0 mL/min at 40 °C. Each fraction was collected every 40 s. Fractions containing *O*-glycopeptide with a branched monosialyl core 1 *O*-glycan were desalted and lyophilized as described above. The lyophilized powder was used as the standard for *O*-glycopeptide with a branched monosialyl core 1 *O*-glycan.

For the preparation of *O*-glycopeptide with a linear monosialyl core 1 *O*-glycan, 10 mL of homogeneous *O*-glycopeptide with a disialyl core 1 *O*-glycan (2 mg/mL) in 50 mmol/L sodium phosphate buffer (pH 5.0) was treated with 0.005 units/mL of neuraminidase (from *Arthrobacter ureafaciens*) at 37 °C for 16 h to release one molecule of Neu5Ac from the *O*-glycopeptide with a disialyl core 1 *O*-glycan. Subsequently, neuraminidase was inactivated via treatment at 100 °C for 5 min. The reaction mixture was loaded onto a C18 RP column equilibrated with 2.0% acetonitrile with 0.1% formic acid. Elution was then performed with a gradient of 2.0–10.8% (0–15 min) and 10.8–90.0% (15–25 min) and an isocratic mode of 90% (25–30 min) at a flow of 10.0 mL/min at 40 °C. Each fraction was collected every 30 s. Fractions containing *O*-glycopeptide with a linear monosialyl core 1 *O*-glycan were lyophilized as described above. The lyophilized sample was used as the standard for *O*-glycopeptide with a linear monosialyl core 1 *O*-glycan.

The amino acid sequence and *O*-glycan structure of homogeneous *O*-glycopeptides, as well as the *O*-glycan binding sites of *O*-glycopeptides with a disialyl or monosialyl core 1 *O*-glycan, were identified through LC–MS analysis as described above. The amount of homogeneous *O*-glycopeptides with a disialyl or monosialyl core 1 *O*-glycan was determined based on the Neu5Ac content. Neu5Ac in the homogeneous *O*-glycopeptides with a disialyl or monosialyl core 1 *O*-glycan was released through neuraminidase treatment, with the released Neu5Ac being determined using HPAE-PAD as described above.

### 2.10. Quantification of Sialyl Core 1 O-Glycans in the MSGP Concentrate

To quantify sialyl core 1 *O*-glycans, 20 μL of the MSGP concentrate (300 μg/mL in H_2_O) or 20 μL of purified homogeneous *O*-glycopeptide (2–100 μg/mL in H_2_O) containing 10 μg/mL of 3′-sialyllewis x (as internal standards) was mixed with 10 μL of Glycan Released Reagent A and 20 μL of Glycan Released Reagent B (provided in the EZGlyco O-glycan Prep Kit) and incubated at 50 °C for 20 min [24]. Subsequently, 100 μL of 1.33 mol/L acetic acid was added for neutralization. The reaction solution was diluted two times with acetonitrile. An aliquot (10 μL) of a released *O*-glycan solution was applied to a Glycanpac AXH-1 column (2.1 mm × 150 mm; 3 μm particle size) and analyzed using a Q-Exactive mass spectrometer coupled with an UltiMate 3000. The mobile phase comprised acetonitrile (solvent A) and 50 mmol/L ammonium formate (pH 4.4) (solvent B) in a gradient elution of 0 to 38.5 min at 10–27.5% B and then 38.5 to 45 min at 27.5% B. The flow rate was 0.4 mL/min. The electrospray voltage and heat capillary temperature were 3.5 kV and 275 °C, respectively. Nitrogen (99.5% purity) was used as sheath gas (set to 35), auxiliary gas (set to 10), and collision gas. Single ion monitoring (SIM) chromatograms were acquired in negative mode at *m*/*z* 489.16 (corresponding to [M−2H]^2−^ of oximated disialyl core 1 *O*-glycan), 688.24 ([M−H]^−^ of oximated monosialyl core 1 *O*-glycans), and 834.30 ([M−H]^−^ of oximated 3′-sialyllewis x). Chromatograms were analyzed using Xcalibur version 4.0 to obtain the SIM area of each sialyl core 1 *O*-glycan. The SIM area of each sialyl *O*-glycan was divided by that of the internal standard, respectively, and each glycan was quantified using a calibration curve generated from that of the purified *O*-glycopeptide.

## 3. Results

### 3.1. Chemical Composition of the MSGP Concentrate

The MSGP concentrate was prepared from commercially available WPI derived from cheese whey. After comparing the gross chemical composition of MSGP concentrate with that of WPI, a two-thirds decrease in protein and a fifteenfold increase in carbohydrates were observed (Table 1). Neu5Ac and GNB (Galβ1-3GalNAc), parts of carbohydrates, were more than 10 times higher in MSGP concentrate than in WPI (Table 1).

### 3.2. SEC Analysis of the MSGP Concentrate

The molecular weight distribution of the MSGP concentrate was analyzed using SEC. The main peak of the MSGP concentrate was observed in the molecular weight range from 1000 to 3500, which was confirmed to have shifted to a lower molecular weight range than that of WPI (Figure 1a,b). The peak top of MSGP (molecular weight: 1399) treated with neuraminidase and *O*-glycosidase capable of releasing core 1 *O*-glycans from glycopeptides shifted to a lower molecular region than that of MSGP (molecular weight: 1904) (Figure 1c). Although MSGP treated with proteinase K showed no change in molecular weight distribution, MSGP treated with proteinase K after treatment with neuraminidase and *O*-glycosidase demonstrated a shift to a much lower molecular region than that of MSGP treated with neuraminidase and *O*-glycosidase (Appendix A), indicating that the main peak of MSGP determined via SEC consisted of glycopeptides with sialyl core 1 *O*-glycans.

### 3.3. LC–MS/MS Analysis of Glycopeptides in the MSGP Concentrate

Glycopeptides in the MSGP concentrate were analyzed using LC–MS/MS with CID and ETD. Mass spectra were examined using Proteome Discoverer (version 2.5), Byonic software (version 2.13.17), and manual inspection. A total of 141 glycopeptides were detected, among which 131 were *O*-glycopeptides derived from the GMP and 10 were *N*- or *O*-glycopeptides derived from osteopontin and glycosylation-dependent cell adhesion molecule 1 (Appendix A). These results demonstrate that GMP-derived *O*-glycopeptides were predominantly enriched in the MSGP concentrate. The 131 types of GMP-derived *O*-glycopeptides comprised combinations of 43 types of peptides and 7 types of glycans (Figure 2). The peptide chain consisted of ≤12 amino acid residues. The disialyl *O*-glycan (glycan **5** in Figure 2B) was the most frequently detected (Appendix A). Parts of the disialyl *O*-glycans were *O*-acetylated at Neu5Ac binding to GalNAc (glycans **6** and **7** in Figure 2B). MS/MS analysis with ETD identified eight amino acid residues as glycosylation sites (^121^T, ^131^T, ^133^T, ^136^T, ^141^S, ^142^T, ^165^T, and ^167^T shown in red letters in Figure 2A; residue numbering is based on Swiss–Prot entries for the mature form of bovine κ-casein; accession number P02668). In addition, the detection of glycosylated EA^149^SPE and A^149^SPE (Nos. 121 and 122 in Appendix A) indicated the glycosylation of ^149^S, whereas the detection of glycosylated ^155^SPPEIN^161^TVQ and ^155^SPPEIN^161^TVQVT (Nos. 123 and 124 in Appendix A) indicated the glycosylation of ^155^S and/or ^161^T.

### 3.4. Composition of O-Glycans in the MSGP Concentrate

LC–FLD was used to analyze the composition of *O*-glycans in the MSGP concentrate after being chemically released and fluorescently (2-AB) labeled using a commercially available kit. The proportion of each 2-AB-labeled *O*-glycan was determined from the signal area of the FL chromatogram (Figure 3). One type of asialyl core 1, three types of sialyl core 1 *O*-glycans, and one type of sialyl core 2 *O*-glycan were detected. The signal area for asialyl core 1 (GNB) could not be evaluated correctly due to overlapping lactose signals. Regarding the proportion of sialyl *O*-glycans excluding peeling by-product in the MSGP concentrate, disialyl core 1 *O*-glycan (glycan **5** in Figure 2B) was the most abundant (72.4%), whereas sialyl core 2 *O*-glycan was the scantest (2.2%) among the sialyl *O*-glycans in the MSGP concentrate (Table 2).

### 3.5. Quantification of Sialyl Core 1 O-Glycan in the MSGP Concentrate

We attempted to quantify sialyl core 1 *O*-glycans (monosialyl and disialyl glycans), which were the most significant glycans in the MSGP concentrate, through eliminative oximation [14] with some modifications [15]. To adjust the efficiency of glycan release and rate of peeling reaction of the released glycans, homogeneous glycopeptides with monosialyl or disialyl core 1 *O*-glycans were prepared as standards. First, homogeneous glycopeptide with disialyl core 1 *O*-glycan was purified from the MSGP concentrate via two-step HPLC purification (Appendix A). The structure of the purified *O*-glycopeptide was determined to be 8-aa *O*-glycopeptide (GEPTSTPT) with a disialyl core 1 *O*-glycan (Neu5Acα2,3Galβ1,3(Neu5Acα2,6)GalNAc; glycan **5** in Figure 2B) based on LC–MS with CID- and ETC-MS/MS analysis (Figure 4A,D). Thereafter, two types of monosialyl glycopeptides were prepared through the digestion of homogeneous glycopeptides with a disialyl core 1 *O*-glycan using α2,3-specific or non-specific neuraminidase and HPLC purification (Appendix A). The structures of the purified monosialyl glycopeptides were determined to be 8-aa *O*-glycopeptide (GEPTSTPT) with a linear (Neu5Acα2,6Galβ1,3GalNAc, glycan **3**; Figure 4B,E) and branched (Galβ1,3(Neu5Acα2,6)GalNAc, glycan **4**; Figure 4C,F) monosialyl core 1 *O*-glycan based on LC–MS. The *O*-glycans in homogeneous glycopeptides and MSGP concentrate were released via eliminative oximation, after which the oximated *O*-glycans were detected using LC–MS without further labeling reaction. The amount of sialyl core 1 *O*-glycans in the MSGP concentrate was calculated using calibration curves generated by homogeneous glycopeptides with sialyl core 1 *O*-glycans (Figure 4G–I). Linear monosialyl *O*-glycan **3**, branched monosialyl *O*-glycan **4**, and disialyl *O*-glycan **5** accounted for 1.3%, 0.4%, and 19.7% (wt/wt) of the MSGP concentrate, respectively. Disialyl *O*-glycan **5** was the major glycan, accounting for 92% of the sialyl core 1 *O*-glycans in the MSGP concentrate.

## 4. Discussion

In the current study, we prepared an MSGP concentrate to enrich the sialyl *O*-glycan moiety found in whey protein and used SEC, LC–FLD, and LC–MS with CID-MS/MS and ETD for both qualitative and quantitative analyses of the same. Notably, SEC and LC–MS analyses revealed that the molecular weight distribution of the sialylglycopeptides contained in the MSGP concentrate ranged from 1000 to 3500. Moreover, LC–FLD determined that the sialyl *O*-glycans in the MSGP concentrate consisted of disialyl core 1 *O*-glycans, linear monosialyl core 1 *O*-glycans, branched monosialyl core 1 *O*-glycans, and sialyl core 2 *O*-glycans. The total amount of the three types of sialyl core 1 *O*-glycans in the MSGP concentrate was 21.4% (wt/wt), with the disialyl core 1 *O*-glycan being the major glycan identified using the eliminative oximation method and a calibration curve generated from homogeneous glycopeptides. Considering their potential to become a new functional material, these sialyl *O*-glycans certainly hold promise for maintaining and improving health. Therefore, controlling for the quality and quantity of these sialyl *O*-glycans is crucial when using the MSGP concentrate as a nutraceutical food. The results of the current study offer a new approach and technology for quality control checks of glycopeptides in the MSGP concentrate to ensure the reliability and safety of this potential health product.

The present study tackled the challenging task of quantifying *O*-glycans accurately. To quantify *O*-glycans, chemically releasing them from glycopeptides is necessary. In general, although mild reaction conditions are often employed to suppress side reactions, this frequently promotes insufficient glycan release and poor quantification. Recently, three groups have described a novel and versatile method for analyzing *O*-glycans through β-elimination in the presence of a pyrazolone analog [25,26,27]. These methods combine release under alkaline conditions and *O*-glycan labeling using 1-phenyl-3-methyl 5-pyrazolone, which minimizes the peeling reaction. Alternatively, Kameyama et al. reported a method for releasing *O*-glycans from glycoproteins using hydroxylamine and an organic superbase (DBU) and tagging of released *O*-glycans using 2-AA [23]. Their method involves reacting hydroxylamine with the released glycans to form oximes, which are relatively stable under alkaline conditions. We selected the eliminative oximation method for the release reaction considering its ability to efficiently release *O*-glycans while suppressing decomposition. In the present study, *O*-glycans were analyzed in the oximated form without any further labeling reaction through LC–MS to avoid loss of quantification due to glycan degradation associated with the labeling reaction [24]. Furthermore, accurate quantification of *O*-glycans requires *O*-glycopeptide standards, which are not commercially available. We addressed this issue by preparing three types of homogeneous *O*-glycopeptide standards with different *O*-glycans from the MSGP concentrate through purification and enzymatic conversion. Ultimately, the combination of the eliminative oximation method and the preparation of *O*-glycopeptide standards allowed for the successful quantification of *O*-glycans. Our results showed that sialyl core 1 *O*-glycans accounted for 21.4% (wt/wt) of the MSGP concentrate. The Neu5Ac content bound to sialyl core 1 *O*-glycans is estimated to be 13.4%, a calculation derived from the molecular weights of the sialyl core 1 *O*-glycans (glycans **3** and **4**: 674.61, glycan **5**: 965.87) and of Neu5Ac (309.27), indicating that around 82% of the total Neu5Ac content (16.4%) contained in the MSGP concentrate is present as a sialyl core 1 *O*-glycan. This result seems reasonable considering that some Neu5Ac molecules in the MSGP concentrate exist as sialyl *N*-glycans and sialyl core 2 *O*-glycans. To the best of our knowledge, this is the first report of the absolute quantification of *O*-glycans. Further studies are definitely required given the constant evolution of methods for quantifying *O*-glycan.

We detected 141 glycopeptides in the MSGP concentrate, among which 131 were *O*-glycopeptides derived from GMP. This result is understandable considering that the WPI used as a raw material contains approximately 15% GMP, the most abundant glycoprotein in WPI. Of these *O*-glycopeptides, 84 *O*-glycopeptides were commonly detected in both the WPI-derived MSGP concentrate in this study and the G-WPC-derived MSGP concentrate in our previous study [21]. The seven types of *O*-glycans were completely identical in both MSGP concentrates. These results demonstrated that an equivalent MSGP concentrate could be prepared from different ingredients. This finding has significant implications for the versatility of the MSGP concentrate preparation process. Interestingly, Kurogochi et al. reported that several types of *O*-glycopeptides derived from GMP can be prepared from trypsin-digested WPC or WPI from cheese whey [5]. *O*-glycopeptides are an attractive material for numerous purposes, such as identifying glycosylation sites using MS and exploring the substrate specificity of endoglycosidases and endoproteases. GMP is a good material for preparing *O*-glycopeptides.

Previous studies have reported that GMP has seven glycosylation sites (^121^T, ^131^T, ^133^T, ^136^T, ^141^S, ^142^T, and ^165^T) [15,16]. In line with this, the current study detected the glycopeptides in which these seven sites were glycosylated, as well as the glycopeptides in which threonine/serine residues other than these seven sites (^149^S, ^155^S/^161^T, and ^167^T) may be glycosylated. Unfortunately, given the low abundance of glycopeptides that could be new glycosylation sites, we were unable to analyze the binding sites through ETD. To confirm whether new binding sites exist, further enrichment of the glycopeptides containing them is necessary. Although several studies have analyzed glycans in GMP over the years, several unknown variables still remain.

The function of the WPI-derived MSGP concentrate has yet to be verified. Koh J. et al. reported that the intact GMP is digested in the human jejunum to produce glycopeptides [28]. The glycopeptides derived from GMP they detected in jejunal fluids are structurally similar to the glycopeptides contained in the MSGP concentrate. If the GMP fragments digested in the jejunum have biological activities [29], then the glycopeptides contained in the MSGP concentrate should also have activities similar to those of the GMP fragments. Further studies are needed to elucidate the function of the MSGP concentrate.

While the physical properties of GMP have been the subject of investigation, as well as its effects on the physical properties of other milk proteins, specific experiments addressing these aspects of the MSGP concentrate have yet to be conducted. A comprehensive understanding of the MSGP concentrate is essential to elucidate its potential utilization in functional foods. This understanding should encompass the stability, solubility, and emulsifying properties of the MSGP concentrate itself, as well as the interaction with other components in the food matrix. To fully grasp the possible applications of the MSGP concentrates, these investigations are included in our future research plan.

Aside from WPI from cheese whey, edible bird’s nest, breast milk, and eggs have been found to be rich in sialic acid. However, the methods presented in the current study cannot be applied to all of these foods. Considering that the structures and amount of sialyl *O*-glycans vary depending on the food, analytical methods suitable for each food may need to be considered.

In conclusion, we performed qualitative and quantitative analyses of sialyl *O*-glycans in the MSGP concentrate prepared through proteolytic digestion and ultrafiltration of WPI. LC–FLD analysis showed that the sialyl *O*-glycans in the MSGP concentrate mainly consisted of three types of sialyl core 1 *O*-glycans. These three *O*-glycans accounted for a total of 21.4% (wt/wt) of the MSGP concentrate, among which the disialyl core 1 *O*-glycan accounted for over 90%. These results revealed that MSGP concentrate is a food material with high sialyl core 1 *O*-glycan content. This study provides new insights into assessment methods for the quality control of *O*-glycan in food products.

## Figures and Tables

**Figure 1 foods-13-02792-f001:**
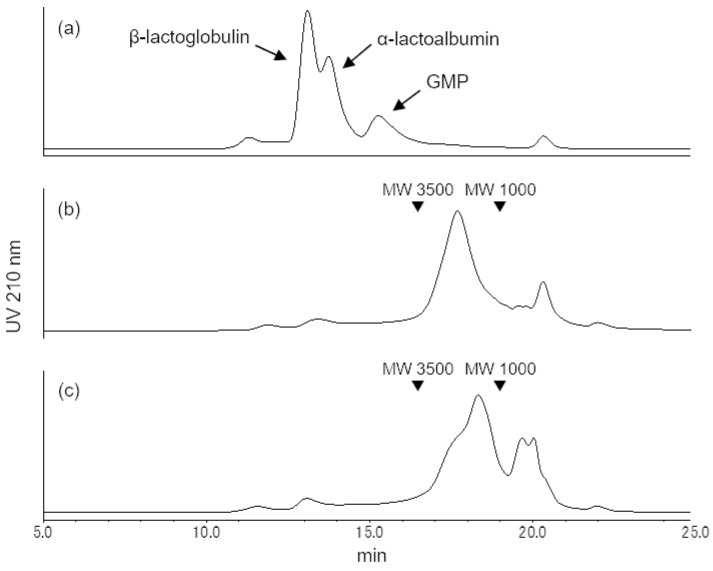
Size-exclusion chromatograms of WPI, MSGP concentrate, and MSGP concentrate treated with glycosidases. WPI (**a**), MSGP concentrate (**b**), MSGP concentrate treated with neuraminidase, and *O*-glycosidase (**c**) were applied to a tandem combination of Inertsil Diol and Inertsil WP300 Diol columns (GL Sciences) and eluted with 40% (vol/vol) acetonitrile containing 0.01% trifluoroacetic acid at a flow rate of 0.3 mL/min. Absorbance was monitored at 210 nm. The inverted triangle symbol indicates the retention time of the molecular weight (MW) calculated from β-lactoglobulin (MW: 18,277), α-lactalbumin (MW: 14,146), aprotinin (MW: 6512), sialylglycopeptide (from egg yolk, MW: 2866), and oxytocin (MW: 1007).

**Figure 2 foods-13-02792-f002:**
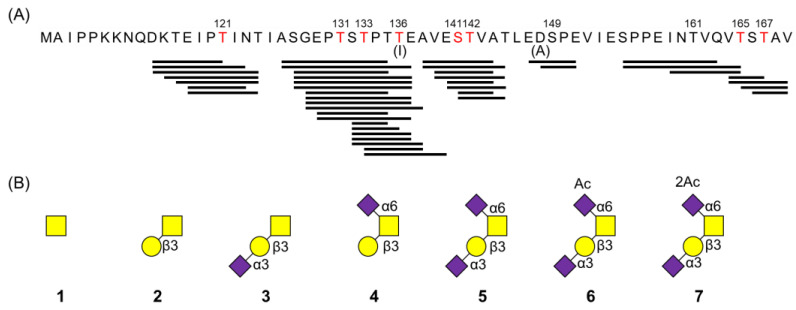
Glycopeptides detected in WPI-derived MSGP concentrate using LC–MS/MS with CID and ETD. (**A**) The sequence of the GMP is shown at the top (residue numbering is based on Swiss–Prot entries for the mature form of bovine κ-casein; accession number P02668). Black bars below the GMP sequence indicate peptides detected via LC–MS, whereas red letters in the GMP sequence indicate glycosylated residues characterized by ETD measurements. (**B**) Proposed structures of glycans. Yellow squares, yellow circles, and purple diamonds indicate GalNAc, galactose (Gal), and Neu5Ac, respectively. *O*-Acetylated Neu5Ac or *O*, *O*’-diacetylated Neu5Ac was represented by the addition of Ac or 2Ac on top of Neu5Ac.

**Figure 3 foods-13-02792-f003:**
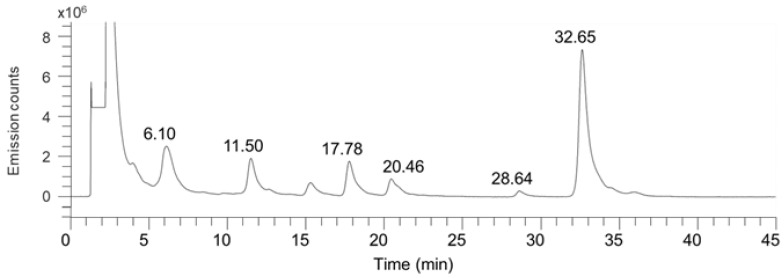
HPLC fluorescence chromatogram of 2-AB-labeled *O*-glycans from the MSGP concentrate. The labeled *O*-glycans were applied to a Glycanpac AXH-1 Column at 40 °C and eluted with a gradient of acetonitrile and 50 mmol/L ammonium formate solution (pH 4.4) in a gradient of 10–35% ammonium formate for 0 to 55 min. The flow rate was maintained at 0.4 mL/min. The excitation and emission wavelengths were set at 330 and 430 nm, respectively. The peaks at 17.78, 20.46, 28.64, and 32.65 min correspond to linear monosialyl core 1 *O*-glycan **3**, branched monosialyl core 1 *O*-glycan **4**, monosialyl core 2 *O*-glycan, and disialyl core 1 *O*-glycan **5**, respectively. The peaks at 6.10 and 11.50 min correspond to GNB overlapped with lactose and peeling by-product, respectively. Each peak was assigned based on MS2 analysis.

**Figure 4 foods-13-02792-f004:**
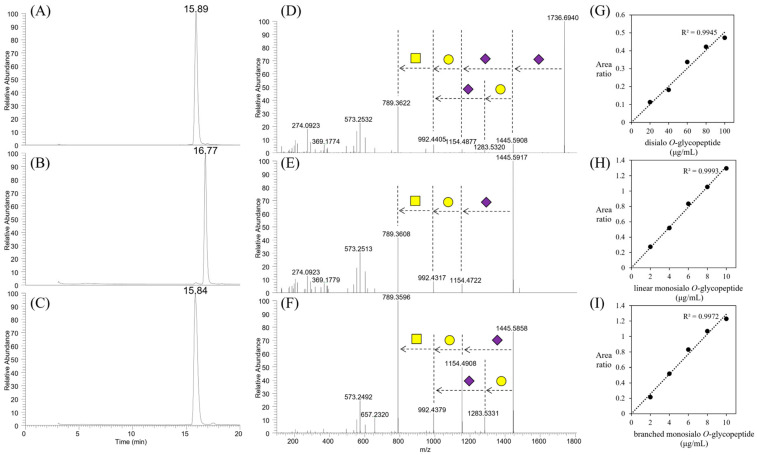
Preparation of homogeneous *O*-glycopeptides with sialyl core 1 *O*-glycans for use as standards. TIC chromatograms (*m*/*z* 400–2000) of the purified glycopeptides with disialyl (**A**), linear monosialyl (**B**), and branched monosialyl (**C**) core 1 *O*-glycans. MS2 spectra of the purified *O*-glycopeptides with disialyl (**D**), linear monosialyl (**E**), and branched monosialyl (**F**) core 1 *O*-glycans. Yellow squares, yellow circles, and purple diamonds indicate GalNAc, Gal, and Neu5Ac, respectively. Calibration curves for the LC–MS area of disialyl (**G**), linear monosialyl (**H**), and branched monosialyl (**I**) *O*-glycans released from the purified *O*-glycopeptides through eliminative oximation.

**Table 1 foods-13-02792-t001:** Chemical specifications of WPI and the MSGP concentrate prepared from WPI (wt/wt).

	WPI	MSGP Concentrate
Moisture	4.5%	2.0%
Protein	90.0%	60.8%
Fat	0.5%	2.9%
Ash	3.0%	4.2%
Carbohydrate	2.0%	30.1%
Neu5Ac	1.4%	16.4%
GNB	0.9%	10.1%

**Table 2 foods-13-02792-t002:** Composition of sialyl *O*-glycans of the MSGP concentrate.

*O*-Glycan	Proportion
Linear monosialyl core 1 *O*-glycan **3**	16.5%
Branched monosialyl core 1 *O*-glycan **4**	8.9%
Disialyl core 1 *O*-glycan **5**	72.4%
Monosialyl core 2 *O*-glycan	2.2%

## Data Availability

The original contributions presented in the study are included in the article/Appendix A, further inquiries can be directed to the corresponding author.

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
