# Peer review of "Qualitative and Quantitative Analyses of Sialyl O-Glycans in Milk-Derived Sialylglycopeptide Concentrate"

_foods, 2024, doi:10.3390/foods13172792_

Round 1

Reviewer 1 Report

Comments and Suggestions for Authors

Remarks

Introduction

Structure and location of glycosidic moieties in kappa-casein chain has beeen discovered before 2000. Please cite original works providing this information. Qualitative part of Author's work only confirms information obtained decades ago.

Information about additional O-acetylation of NeuAc in kappa-casein sugar moieties has been presented by some of Authors in reference cited as No 5. Is it first information about this modification of kappa-casein ugar moieties? If yes - why not cite it in this context? If not, please cite original work.

Please add some background concerning significance of carbohydrate moieties, especially sialic acid for biological and technological properties of kappa casein and its fragments. There are e.g. recent review articles concerning possible health-promoting role of dietary sialic acid.

Please add background concerning analysis of glycosylated macropeptide fragments using HPLC and mass spectrometry.

Materials and methods

Please add EC numbers of all enzymes used in experiment. Please define units of activity of these enzymes.

Is it possible to explain acetonitile concentration in mobile phase used for SEC experiment. There are publications recommending higher concentration, up to 70% ACN. Wht was reason of use of very low concentration of TFA in mobile phase for SEC?

Results 

Legend to the figure 2.

Please explain abbreviation Ac in legend to the figure.

Discussion

Results should be discussed using historical background. Including structure and location glycosidic moieties and previous attempts of analysis of macropeptide fragments containing sugar residues. There are publications concerning this topic, not cited in manuscript.

On this basis Authors should more clearly point out what is really new in Their work.

There are some results indicating that glycosylated macropeptide (especially with high content of carbohydrates) is less suceptible to proteolysis than sugar-free macropeptide. Do Authors confirm these results? If not please state it. Please discuss this point in the context of use CMP-derived glycopeptides as potential components of functional food.

Reviewer 2 Report

Comments and Suggestions for Authors

This manuscript qualitatively and quantitatively analyzes the presence of the sialyl O-glycans in MSGP concentrate, which is a meaningful proposal, but there are still some minor issues:

1. There are minor grammar errors in the manuscript, please carefully check.

2. It is recommended to add relevant content on qualitative and quantitative methods in the Introduction.

3. It is suggested that the author add references in the methodology and discussion section.

4. It is suggested that the author further investigate the stability, solubility, and emulsifying properties of the separated substance.

5. What is the relative molecular weight of the substance separated by the author?
